# Broad Adversarial Training with Data Augmentation in the Output Space

**Nils Worzyk**  **Stella X. Yu**

UC Berkeley / International Computer Science Institute, Berkeley, CA, USA
nilswo@icsi.berkeley.edu,   stellayu@berkeley.edu

## Abstract

In image classification, data augmentation and the usage of additional data has been shown to increase the efficiency of clean training and the accuracy of the resulting model. However, this does not prevent models from being fooled by adversarial manipulations. To increase the robustness, Adversarial Training (AT) is an easy, yet effective and widely used method to harden neural networks against adversarial inputs. Still, AT is computationally expensive and only creates one adversarial input per sample of the current batch. We propose Broad Adversarial Training (BAT), which combines adversarial training and data augmentation in the decision space, i.e., on the models output vector. By adding random noise to the original adversarial output vector, we create multiple pseudo adversarial instances, thus increasing the data pool for adversarial training. We show that this general idea is applicable to two different learning paradigms, i.e., supervised and self-supervised learning. Using BAT instead of AT for supervised learning, we can increase the robustness by 0.56% for small seen attacks. For medium and larger seen attacks, the robustness increases by 4.57% and 1.11%, respectively. On large unseen attack, we can also report an increase in the robustness by 1.11% and 0.29%. When combining a larger corpus of input data with our proposed method, we report a slight increase of the clean accuracy and increased robustness against all observed attacks, compared to AT. In self-supervised training, we monitor a similar increase in robust accuracy for seen attacks and large unseen attacks, when it comes to the downstream task of image classification. In addition, for both observed self-supervised models, the clean accuracy also increases by up to 1.37% using our method.

## Introduction

The performance of deep learning models in various domains, e.g., image classification (Zhai et al. 2021), semantic image segmentation (Tao, Sapra, and Catanzaro 2020), or reinforcement learning (Tang et al. 2017) is already on a high level and constantly improving. Among other aspects, ongoing research and advances in data augmentation (Cubuk et al. 2020) techniques, as well as the creation of more realistic synthetic inputs (Ho, Jain, and Abbeel 2020) contribute to this success. Both techniques aim to enrich the training data, which increases the performance. However, when it

comes to safety-critical applications, e.g., autonomous driving, adversarial inputs pose a threat. By applying small but malicious manipulations to the input, the prediction of the model can change drastically.

Starting by manipulating digital inputs, several authors, e.g., (Goodfellow, Shlens, and Szegedy 2014; Carlini and Wagner 2017; Madry et al. 2017), developed different techniques to calculate and create the necessary manipulations to fool neural networks into misclassifying a given input. Later, these attacks were adapted or extended to also work in the physical world (Athalye et al. 2018; Worzyk, Kahlen, and Kramer 2019; Ranjan et al. 2019).

One widely used technique to harden neural networks against such attacks is Adversarial Training (AT), which is simple but yet very effective. The idea is to create adversarial inputs during the training process, and include or exclusively use them for training. Thereby, the model learns to be more resilient against these worst-case perturbations. (Madry et al. 2017) for example, proposed a method referred to as Projected Gradient Descent (PGD) which is very successful in finding adversarial inputs and furthermore use these adversarial instances exclusively for training a given model. Even though effective, all to us known adversarial training techniques, only create one adversarial input per sample in the current batch.

To increase the impact of any given adversarial instance during adversarial training, we propose to combine adversarial training and data augmentation in the decision space, specifically, manipulating the output vector of a given adversarial instance. In contrast to clean inputs, which represent the assumed reality in the input space, adversarial instances are calculated, and basically exist, based on flaws in the decision boundary of a model. Ultimately, the output vector of a given model defines the decision calculated by the model. Similar to data augmentation on clean inputs increasing the data pool and leading to better clean performance, increasing the data pool of adversarial samples in the output space leads to better robustness. The overall concept is shown in Figure 1.

In Figure 1a, the decision space during traditional adversarial training is displayed. Based on the current decision boundary (bold line) and the output vector for a clean sample (orange circle), an adversarial input (blue cross) is created, whose output vector is located on the *wrong* side of the de-

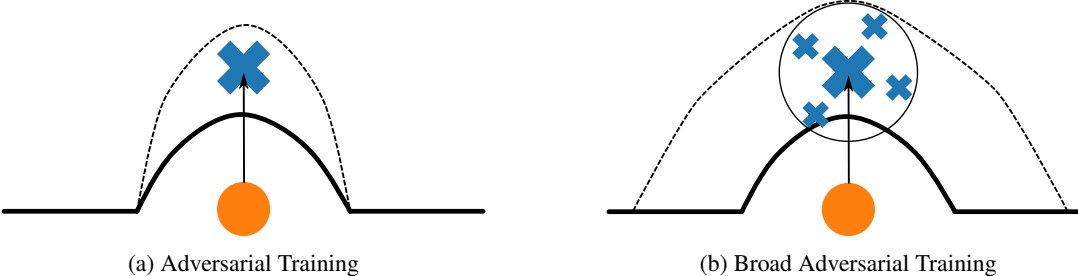

● Clean Sample    ✖ Adversarial Sample    — Clean decision boundary    --- Adversarial decision boundary

(a) Adversarial Training                          (b) Broad Adversarial Training

Figure 1: Difference between traditional (1a) and Broad (1b) Adversarial Training. Given the output vector of a clean sample (orange circle) and the current decision boundary (solid line), adversarial inputs (blue cross) lying on the *wrong* side of the decision boundary are created. When adding the adversarial samples to the training process, the decision boundary adapts accordingly (dashed line). During Broad Adversarial Training, the initial adversarial output vector is perturbed randomly within a given radius to create a set of additional pseudo adversarial inputs (smaller blue crosses). This extends the impact of any single adversarial input on the adversarial decision boundary (dashed line) during training.

cision boundary. The adversarial decision boundary (dashed line) is then optimized to contribute for the adversarial input.

Figure 1b outlines our extension to this process. Based on the output vector of an adversarial input (large blue cross), multiple pseudo adversarial inputs (small blue crosses) are created by applying conditioned normally distributed random noise within a predefined radius to the initial adversarial output vector. By *scattering* the adversarial output vector, we widen its impact on the new adversarial decision boundary (dashed line).

The remainder of this paper is structured as follows. In the Background section, we introduce the supervised, as well as self-supervised adversarial training methods used for the experiments in this paper. Afterwards, we define our proposed BAT method in more detail, followed by the experiments and their discussion. Finally, we put our work in context with other related work, and conclude the paper.

## Background

### Supervised Adversarial Training

**PGD:** One of the widest known techniques for adversarial training is Projected Gradient Descent (PGD), proposed by (Madry et al. 2017). Instead of using clean samples during training, the authors use the corresponding adversarial instances, created based on the following iterative equation,

$$x^0 = x$$
$$x^{i+1} = \Pi_{B(x,\varepsilon)}\left(x^i + \alpha\text{sign}\left(\nabla_{x^i}\mathcal{L}_{\text{CE}}\left(\theta, x^i, y\right)\right)\right) . \tag{1}$$

This function is initialized by the original input $x$ and calculates the gradients, regarding the intermediate adversarial input $x^i$, of the cross-entropy loss $\mathcal{L}_{\text{CE}}$ away from the true label $y$, based on the current parameters of the model $\theta$. The sign of the gradients is multiplied by a step size parameter $\alpha$ and added to the current intermediate adversarial input. The projection $\Pi$ then limits the perturbation to be within an $\varepsilon$-ball around the initial input $x$. The parameter $\varepsilon$ essentially governs the allowed amount of perturbation. For example,

$\varepsilon = 8/255$ regarding the $\ell_\infty$ norm would state, that each pixel of the original input vector is allowed to be increased or decreased by not more than 8 values.

### Self-supervised Adversarial Training

More recently, adversarial training is also applied to self-supervised training. The general goal of self-supervision is to train for some pretext tasks where no labels are required. After training, the model and its parameters are transferred to a given downstream task, e.g., image classification. Therefore, the overall model in self-supervised learning is split into a backbone and a projector. The backbone can be based on, e.g., a ResNet architecture (He et al. 2016), stripped of the last fully connected layer. The projector reduces the dimensionality of the backbones output vector to a usually 128-dimensional vector. To train without labels, given a batch of samples $\{x_1, ..., x_b\}$, each sample is duplicated and transformed by a given series of random transformations $t$, e.g., cropping and flipping. The resulting transformed versions of the same origin, i.e., $t_1(x_i)$ and $t_2(x_i)$ are called positive pair, while pairs of samples with different origin, i.e., $t(x_i)$ and $t(x_j)$ with $i \neq j$ are called negative pair. The pretext task of the models used in this paper is to maximise the distance between the output vectors of negative pairs while minimising the distance between the output vectors of positive pairs. How the distance is calculated differs between the training techniques.

After pretraining for the pretext task, the backbone is kept and the projector is discarded. Instead of the projector, a downstream task-specific head is attached. The parameters of the backbone are usually frozen and only the head is trained. In essence, self-supervised pretraining aims to learn good feature representations which can, later on, be used for the given downstream task.

**RoCL:** (Kim, Tack, and Hwang 2020) added adversarial training to the SimCLR (Chen et al. 2020) framework and dubbed it Robust Contrastive Learning (RoCL). To calculate the distance between positive and negative pairs, SimCLR

uses the cosine similarity sim. Because of the different loss function, (Kim, Tack, and Hwang 2020) adapted PGD (cf. Equation 1) accordingly to use the contrastive loss instead of the cross-entropy loss. All functions are given in Table 3 in Appendix .

To implement adversarial training, (Kim, Tack, and Hwang 2020) essentially use the adversarial inputs to extend the positive and negative pairs to triplets. During training, they aim to minimize the distance between the two transformed inputs of the positive pairs, as well as the distances between the two transformed inputs each and the thereof created adversarial input. While maximising the distance to the negative samples. The formalization of the RoCL objective is given in Table 3, where $t(x) + \delta$ is the adversarial sample. The overall training loss is then calculated based on the standard contrastive loss only considering the clean transformed samples, plus the adversarial loss based on the triplets of two transformed inputs and the additional adversarial input.

One challenge using SimCLR as the basic framework is that it requires a large batch size to achieve good performance (Chen et al. 2020). In this type of self-supervised learning, the number of observed negative samples is essential for a good performance, and SimCLR does not incorporate any form of dictionary or memory bank to increase their number. Only the samples from the current batch are used for the calculations. When including adversarial samples into the training process, the number of inputs to be stored on the GPU is increased and results in a reduction of the feasible batch size.

**AMOC:** Another widely known self-supervised framework is Momentum Contrast (MoCo) proposed by (He et al. 2020). The conceptual idea is the same as for SimCLR, i.e., minimising the distance between positive instances while maximising the distance towards negative samples. However, to overcome the problem of large batch sizes, (He et al. 2020) implement a dictionary or memory bank. In addition, they use two networks of the same architecture and the same initial weights. One model is referred to as the query encoder, which is updated after each batch as usual. The other model is called momentum or key encoder, whose parameters are a copy of the query encoder, delayed by a predefined momentum. This makes the output of the key encoder slightly different from the output of the query encoder, which can be considered as an additional form of data augmentation. After processing the inputs of the current batch by both models, the output vectors of the momentum encoder are enriched by output vectors of previous batches, stored in the dictionary. Thereby, a large number of negative samples can be created consistently, leading to overall better performance. The loss, given in Table 3, is then calculated based on the output vectors $q$ of the query encoder and the enriched output vectors $k$ of the key encoder. Furthermore, $\tau$ is a temperature parameter, and $\mathcal{M}$ refers to the memory bank of old key vectors.

Based on this framework, (Xu and Yang 2020) proposed an extension for adversarial training. They introduce a second memory bank to store exclusively the historic adversarial output vectors, and to further disentangle the clean and adversarial distribution, they use dual Batch Normalization as proposed by (Xie et al. 2020). The optimization problem they solve is given in Table 3, with $t_1$ and $t_2$ being two different random transformations from a set $\mathcal{T}$ of possible transformation, and $\delta$ being the adversarial perturbation. $\mathcal{M}_{\mathrm{clean}}$ and $\mathcal{M}_{\mathrm{adv}}$ refer to the clean, and adversarial memory bank, respectively. As for the loss function, (Xu and Yang 2020) tested different memory bank and batch normalization combinations, and reported good results for a combination they refer to as ACC. The 'A' indicates, that the adversarial perturbation is injected into the query encoder, while the key encoder does not observe and perturbation, as well as the clean memory bank $\mathcal{M}_{\mathrm{clean}}$ is used. The formulation to calculate the ACC loss is given in Table 3. Intuitively, by comparing the adversarial output vectors of the query encoder with clean samples from the key encoder, as well as the memory bank, the query encoder $f_q$ learns to classify adversarial inputs as its clean augmentation. To create the adversarial perturbation in the first place, (Xu and Yang 2020) use PGD as well, but with the MoCo loss instead of the cross-entropy loss. Finally, the overall training loss is calculated as a weighted sum of the standard MoCo loss solely trained on clean data, and the selected, e.g., ACC loss to incorporate adversarial instances.

## Method

Multiple approaches for adversarial training are outlined in the Background section. Our goal is to develop a method that is not specifically tailored to one approach, but rather generalizable between different sorts of adversarial training. Therefore, given an input $x$ and a neural network $f$, $f(x)$ denotes the general output vector. In supervised learning, this vector would be the logits, while in self-supervised learning, this would be the 128-dimensional output vector of the projector.

After an adversarial input $x'$ and in consequence also its output vector $f(x')$ has been created by one of the approaches in the Background section, we create multiple pseudo adversarial inputs $f(x'_s)$ by adding conditioned normal distributed random noise,

$$f(x')_s = f(x') + \mathcal{N}(0,1) \cdot \delta_{x',x} \cdot \alpha_s , \qquad (2)$$

where $\delta_{x',x}$ is defined as

$$\delta_{x',x} = f(x') - f(x) \qquad (3)$$

and $\alpha_s$ is a hyperparameter, used to scale the ball around the initial adversarial output vector $f(x')$, introduced by the random noise. Intuitively, $\delta_{x',x}$ defines the element-wise difference that the initial adversarial output vector is moved away from the original instance in the decision space, while $\alpha_s$ scales this initial manipulation.

To confirm that this type of decision space data manipulation is suitable and label preserving, we create randomly perturbed adversarial output vectors for a standard, clean trained model (ST) and track their classification behaviour. In Figure 2, the results for an ST model are shown in the most left bar of each group. The blue (bottom) portion of the bar indicates the percentage of pseudo adversarial instances

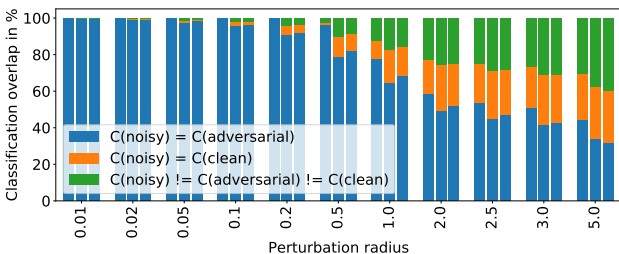

Figure 2: Percentages of pseudo adversarial inputs being classified as indicated, depending on the perturbation scaling factor $\alpha_s$. The bars of each group show the results based on the following models: **Left bar:** Standard trained model; **Middle bar:** Adversarial trained model; **Right bar:** Broad adversarial trained model. C(noisy) = C(adversarial) indicates the perturbed adversarial output vector is classified the same, as the initial adversarial input. C(noisy) = C(clean) gives the percentage of pseudo adversarial inputs, which return to the original true classification area, while C(noisy) != C(adversarial) != C(clean) gives the percentage of pseudo adversarial inputs moving to some different, third class when perturbed randomly.

being classified the same, as the initial adversarial input. The orange (middle) portion indicates the number of pseudo adversarial output vectors returning to the classification area of the initial clean sample, and the green (top) portion gives the percentage of instances that move to a third classification area, which is neither the class of the clean nor the initial adversarial sample.

We can observe that for sufficiently small perturbation 100% of the pseudo adversarial instances are classified the same, as the initial adversarial input. This demonstrates empirically, that the applied conditioned random noise as a form of data augmentation in the decision space can be completely 'label preserving'. Only with larger perturbation radius, more and more perturbed adversarial output vectors move towards a third classification area. The samples returning to their originally true class, however, can be ignored, since the adversarial instances are labelled to have the same class as their clean counterparts during training. Therefore, the assigned label for these instances would not change.

A different perspective to the classification changes shown in Figure 2 is to empirically evaluate the local smoothness of the decision surface. If already for small random perturbation an instance moves into another classification area, the decision boundary might be sharply twisted at that point. If only at larger perturbations the instances move into another class area, the decision boundary can be assumed to be more smooth.

The second bar of each group displays the corresponding behaviour for an adversarially trained model. Similarly to the clean model, at small perturbation radius, almost all pseudo adversarial instances are classified the same as the initial adversarial instance. However, with increasing manipulation, more and more noisy instances move to the original or a third classification area. Compared to ST, the number

of pseudo adversarial instances staying adversarial reduces. This can be explained by the fact, that the attack strength is kept constant, while the decision boundary in AT is pushed towards the observed adversarial instances. Thereby, in ST the adversarial instances are moved further into the wrong classification area, and thereby can endure more random perturbation before moving either back or to a third classification area. For an AT trained model, the adversarial instances are already closer to the decision boundary and are thereby moved to either the original or a third classification area at lower random perturbation sizes. This observation also indicates, that for training with pseudo adversarial output vectors, the scatter radius should be reduced over time. Thereby, the risk of assigning instances to a third classification area with a potentially incorrect label could be minimized.

As a final comparison, the third bar of each group shows the corresponding classifications for pseudo adversarial instances on a BAT model. Here we can see that the number of pseudo adversarial instances being classified as the initial adversarial instance is higher compared to normal adversarial training. The number of instances moving to a third classification area is smaller as well for the BAT model compared to the AT model. This indicates a smoother local decision boundary when a model is trained with BAT compared to AT.

Having verified that applying random perturbation as a form of data augmentation in the decision space is a valid option, our overall pseudo-code is given in Algorithm 1. Aside from the scalar for the perturbation radius $\alpha_s$, we also introduce a hyperparameter to define the number of additionally created pseudo adversarial instances $s_k$. Each additional pseudo adversarial instance only requires the calculation of random noise and evaluation of the given loss function. Addition and multiplication to create the pseudo adversarial instances regarding time complexity are in $O(1)$, while evaluating the loss function, since independent from the parameters added for BAT, can also be considered to be in $O(1)$. Therefore, our extension to implement BAT adds a time complexity in $O(n)$ with the number of created pseudo adversarial inputs to the overall training procedure. In Table 6 in Appendix , the additional time demand for each scattered input during the different training methods, is empirically evaluated and listed.

To even out the effect of having multiple pseudo adversarial instances, we calculate the mean loss and add it, weighted by some factor $\lambda$, to calculate the overall loss as

$$\mathcal{L}_{\text{total}} = \iota \mathcal{L}_{\text{clean}} + \kappa \mathcal{L}_{\text{adv}} + \lambda \mathcal{L}_{\text{scatter}}, \tag{4}$$

where $\iota, \kappa$, and $\lambda$ could be different weights for the different loss functions.

## Results and Discussion

**Dataset and Model:** All experiments were run on the Cifar-10 dataset (Krizhevsky, Hinton et al. 2009). For supervised learning, we did an additional set of experiments marked with $^+$, which uses another 1 million synthetic data points based on Cifar-10, provided by (Gowal et al. 2021). The authors report an increase in adversarial robustness using the additional synthetic data. The model used for all ex-

**Algorithm 1:** Broad Adversarial Training (BAT).

---

**Input:** Dataset $D$, model $f$, parameter $\theta$, Loss function $\mathcal{L}$, # attack steps $k$, # scatter instances $s_k$, scatter scalar $\alpha_s$

**foreach** *iter* $\in$ number of training iteration **do**

    **foreach** $x \in$ minibatch $B = \{x_1, \ldots, x_m\}$ **do**

        $\mathcal{L}_{\text{clean}} = \mathcal{L}\left(f\left(x\right)\right)$

        $x' = \text{generateAdversarial}\left(x\right)$

        $\mathcal{L}_{\text{adv}} = \mathcal{L}\left(f\left(x'\right)\right)$

        **Broad Adversarial Operation:**

        $\delta_{x',x} = f\left(x'\right) - f\left(x\right)$

        **for** $s_k$ instances **do**

            $f\left(x'\right)_s = f\left(x'\right) + \mathcal{N}\left(0,1\right) \cdot \delta_{x',x} \cdot \alpha_s$

            $\mathcal{L}_s \mathrel{+}= \mathcal{L}\left(f\left(x'\right)_s\right)$

        **end**

        $\mathcal{L}_{\text{scatter}} = \frac{\mathcal{L}_s}{s_k}$

        $\mathcal{L}_{\text{total}} = \iota\mathcal{L}_{\text{clean}} + \kappa\mathcal{L}_{\text{adv}} + \lambda\mathcal{L}_{\text{scatter}}$

        Optimize $\theta$ over $\mathcal{L}_{\text{total}}$

    **end**

**end**

---

periments is a ResNet-18 architecture, implemented in the provided repositories of (Kim, Tack, and Hwang 2020) for RoCL, and (Xu and Yang 2020) for AMOC. The experiments for the supervised case were run based on the AMOC framework. More details are provided in Appendix .

**Hyperparameters for training:** For all experiments, we used the provided hyperparameters suggested by (Kim, Tack, and Hwang 2020) for RoCL, and (Xu and Yang 2020) for AMOC, when applicable. More details are provided in Appendix .

**Attacks:** During training, the adversarial inputs were created governed by a perturbation size of $\varepsilon = 8/255$ regarding $\ell_\infty$. Therefore, the $\ell_\infty$ attacks are referred to as seen, even if only for a small perturbation size, while the $\ell_2$ and $\ell_1$ attacks were completely unseen during the training procedure. For adversarial training, we used the parameters provided by the respective frameworks, listed in Appendix .

To challenge the trained models, the adversarial inputs were created with PGD as given in Equation 1 over 20 iteration steps, with a relative step size $\alpha$ of 0.1 to the given allowed amount of perturbation. The overall evaluation was conducted based on the respective functions in the AMOC framework, which itself draws the attacks from the foolbox framework (Rauber, Brendel, and Bethge 2017).

**Hyperparameters for BAT:** For BAT, we found that a good number of additional inputs is $s_k = 10$. In preliminary studies we found that introducing too many additional data points adds too much noise to the training process and thereby reduces the overall performance. On the other hand, too few pseudo adversarial instances do not have any impact on the overall performance. Similarly, setting the scatter radius too small results in no effect on the results, while setting it too large, as shown in Figure 2, will move the pseudo adversarial inputs increasingly towards and over the decision

boundary of a different classification area, and thereby reduces the performance. For supervised BAT, we found that a surprisingly large initial $\alpha_s = 2.5$ decayed by a cosine scheduler, yields the best results. Training AMOC, setting the initial $\alpha_s = 0.25$ decayed by a cosine scheduler works best, respectively for RoCL an initial $\alpha_{0.1}$ decayed by a stepwise function reducing the initial $\alpha_s$ by 0.01 every 100 epochs.

For the weight of the scatter loss to the overall loss, we found that in supervised broad adversarial training the same weight for the original adversarial loss and the scatter loss works best. Similarly, while pretraining AMOC, an equal contribution of the clean, the original adversarial, and the scatter loss yields the best results. For RoCL, a weight of $\lambda = 0.25$ for the scatter loss yields the best results, combined with a weight of $\iota = \kappa = 1.0$ for the clean and original adversarial loss.

## Results

The results for the supervised experiments are given in Table 1, where each value represents the mean value over 5 different runs. For the self-supervised experiments, the results are listed in Table 2. The upper part reports the results where only the classification head was optimized, while the parameters of the pretrained model were frozen. The lower part, indicated by *Self-supervised + finetune*, reports the results where also the parameters of the pretrained model were optimized during training of the classification head. A **B-** in front of the given method indicates, that our proposed adaptation was applied to the following training mechanism. The results for experiments run for 200 epochs are also the mean value over 5 different runs.

## Discussion

Taking a look at the results of the supervised methods in Table 1, using only the original Cifar-10 data, we can report, that the robust classification accuracy can be increased for all seen, as well as large unseen attack, when using BAT instead of AT. For small seen perturbation, the robust classification accuracy increases by 0.56%, while for large perturbation size the accuracy increases by 1.11%. Considering unseen attack, the robust classification accuracy for small attacks is reduced when using BAT, however, with increasing attack size, this reduction is converted to an increase for large perturbation sizes. Considering $\ell_2$ governed attacks with $\varepsilon = 0.75$, the robust classification accuracy can be increased by 1.11% using BAT, while for $\ell_1$ governed attacks with $\varepsilon = 16.16$, the robust classification accuracy increases only slight by 0.29% using BAT instead of AT.

When using the additional 1 million data points, we can reaffirm that it increases the clean, as well as robust accuracy for all training methods and attacks compared to training without the additional data, as (Gowal et al. 2021) reported. Comparing AT and BAT using additional data, we can report that BAT improves on the robust classification accuracy in all observed attacks, as well as a slight increase in the clean accuracy, compared to AT. Even for small unseen attacks, e.g., $\ell_2$ governed attacks with $\varepsilon = 0.25$, the robust accuracy increases by 0.11% using BAT over AT. For larger

| Method | $A_{nat}$ | seen | | | unseen | | | | | |
| | | PGD20 $l_\infty$ | | | PGD20 $l_2$ | | | PGD20 $l_1$ | | |
| | | $\epsilon$ 8/255 | 16/255 | 32/255 | 0.25 | 0.5 | 0.75 | 7.84 | 12 | 16.16 |
|---|---|---|---|---|---|---|---|---|---|---|
| $\mathcal{L}_{CE}$ | **93.92** | 0.00 | 0.00 | 0.00 | 8.27 | 0.17 | 0.00 | 15.07 | 3.37 | 0.61 |
| AT | 81.85 | 52.49 | 22.21 | 1.25 | **73.83** | **63.11** | 50.91 | **70.52** | **62.95** | 54.66 |
| **BAT** | 76.60 | **53.05** | **26.78** | **2.36** | 69.63 | 61.56 | **52.02** | 67.04 | 61.37 | **54.95** |
| $\mathcal{L}_{CE}^{+}$ | **95.04** | 0.00 | 0.00 | 0.00 | 12.42 | 0.58 | 0.04 | 21.75 | 6.21 | 1.84 |
| $AT^{+}$ | 84.15 | 59.22 | 29.70 | 2.60 | 76.78 | 67.51 | 56.09 | 73.47 | 66.06 | 58.05 |
| $BAT^{+}$ | **84.20** | **59.80** | **30.49** | **2.81** | **76.89** | **68.08** | **56.86** | **73.61** | **66.75** | **58.81** |

Table 1: Results on Cifar-10 for supervised trained models with standard cross entropy training $\mathcal{L}_{CE}$, adversarial PGD training (AT), and our proposed Broad Adversarial Training (BAT). For the experiments marked with $^+$, 1 million additional synthetic data points based on Cifar-10 were used for training. During training, the initial adversarial instances were created governed by $\ell_\infty$ with a strength of $8/255$. All experiments were run 5 times and the mean value is reported.

attacks, the robust accuracy benefits more from using BAT over AT.

Observing the results for AMOC when only the classification head is trained, given in Table 2, we can report similar behaviour. The clean accuracy is slightly reduced, while the classification accuracy for seen attacks increases in all combinations of AMOC and head training, except one combination for a large perturbation size. The increase in robustness can range from 0.03% to 1.01%, depending on the attack size. When AMOC is trained for 1000 epochs, instead of 200, the robust classification accuracy for large and sometimes medium unseen attacks increases as well, between 0.04% and 1.22%.

For RoCL, introducing our proposed pseudo adversarial instances into the self-supervised pretraining, the clean accuracy increases between 0.04% and 1.37%. Also, the robustness against seen attacks increases for small and medium-sized attacks between 0.81% and 2.02%. Interestingly, the robustness for large seen attacks only increases by 0.13% using B-RoCL during pretraining and BAT for the classification head is applied. Similar to AMOC, RoCL also becomes more robust to medium and/or large unseen attacks, when trained with additional pseudo adversarial inputs. The robustness there increases between 0.25% and 3.25%. Particularly for the combination B-RoCL+AT, our proposed pretraining leads to better clean and robustness accuracy against almost all attacks compared to standard RoCL+AT.

When during training of the classification head also the parameters of the pretrained models are finetuned, we observe an increase in clean, as well as robust accuracy for AMOC, too. In particular, comparing B-AMOC+B-AF with AMOC+AF trained for 1000 epochs, we observe that the performance increases against almost all attacks between 0.35% and 0.9%. If we assume AMOC+AF as the reported baseline, B-AMOC+B-AF increase the robustness against all seen attacks between 0.13% and 0.25%, as well as against medium and large unseen attacks between 0.11% and 0.24%.

To further investigate why BAT is sometimes weaker regarding unseen attacks, we calculated the perturbation size of successful $\ell_2$ and $\ell_1$ governed attacks regarding $\ell_\infty$. The resulting distributions are given in Figure 3 in Appendix , where the x-axis indicates the perturbation size regarding $\ell_\infty$, and the y-axis shows the number of successful attacks. We recommend inspecting the figures digitally to zoom in for better visibility. The distribution of manipulation sizes based on attacks controlled by $\ell_2$ is given in blue (legend top), while the values for $\ell_1$-attacks are shown in orange (legend middle), and for $\ell_\infty$-attacks in green (legend bottom). The grey vertical line gives a landmark of a perturbation of $\ell_\infty = 8/255$, which is the perturbation size seen during adversarial and broad adversarial training. The left column of each pair shows the corresponding distributions for small perturbation size, while the right column shows the respective distribution for large perturbation size.

The top row shows the results when the attacked model was trained on clean data only. We can see that the applied manipulation of attacks governed by $\ell_2$ and $\ell_1$ is generally lower than the adversarial manipulation applied by the corresponding $\ell_\infty$-attack. This could explain why even models trained on clean samples are, to some extend, robust against $\ell_2$ and $\ell_1$ controlled attacks, as we can observe in Table 1.

The second and third rows show the resulting perturbation size distributions for attacks on an adversarial trained network, resp. broad adversarial trained model. Here we can see that the perturbation of $\ell_2$- and $\ell_1$-attacks is larger regarding $\ell_\infty$ than the perturbation of the corresponding $\ell_\infty$-attack, especially for a small perturbation size. Since during training both models have seen adversarial samples of the perturbation size $\ell_\infty = 8/255$, this indicates why both also become more robust, but not perfect, against $\ell_2$- and $\ell_1$-attacks in general, but probably not why BAT performs worse than standard AT on unseen attacks.

Observing the pixel level manipulations applied by $\ell_2$- and $\ell_1$-attacks, evaluated regarding $\ell_\infty$ might give more insights why BAT is worse regarding small perturbations, compared to AT. The resulting perturbations, exemplary for the blue color channel of an observed input, are shown in Figure 4 in Appendix for a standard trained model, in Figure 5 for an AT trained model, and in Figure 6 for a broad adversarial trained model. The visualization indicates whether the pixel value of the adversarial input was increased (red, top of the right bar aside the grid) or decreased (blue, bot-

|  |  | seen | | | unseen | | | | | |
| Method | $A_{nat}$ | PGD20 $l_\infty$ | | | PGD20 $l_2$ | | | PGD20 $l_1$ | | |
|  |  | $\epsilon$ 8/255 | 16/255 | 32/255 | 0.25 | 0.5 | 0.75 | 7.84 | 12 | 16.16 |
| --- | --- | --- | --- | --- | --- | --- | --- | --- | --- | --- |
| *Self-supervised:* | | | | | | | | | | |
| 200 epochs: | | | | | | | | | | |
| AMOC + $\mathcal{L}_{CE}$ | **79.03** | 36.61 | 7.46 | 0.05 | **67.93** | **54.82** | **41.53** | **66.27** | **58.53** | **50.74** |
| **B-AMOC** + $\mathcal{L}_{CE}$ | 78.88 | **37.09** | **8.15** | 0.05 | 67.64 | 54.58 | 41.37 | 65.77 | 57.91 | 50.19 |
| AMOC + AT | **74.79** | 43.97 | 14.53 | 0.19 | **67.10** | **58.10** | 48.09 | **66.03** | **60.78** | **54.92** |
| **B-AMOC** + AT | 74.58 | **44.57** | **15.45** | **0.26** | 66.88 | 57.93 | **48.21** | 65.72 | 60.43 | 54.64 |
| AMOC + **BAT** | **74.32** | 44.08 | 15.15 | 0.24 | 66.63 | **57.92** | **48.21** | 65.62 | **60.78** | **54.82** |
| **B-AMOC** + **BAT** | 74.25 | **44.59** | **15.85** | **0.28** | **66.64** | 57.75 | 48.18 | 65.49 | 60.21 | 54.44 |
| 1000 epochs: | | | | | | | | | | |
| AMOC + $\mathcal{L}_{CE}$ | **86.52** | 44.91 | 11.46 | 0.11 | **77.04** | 63.59 | 50.39 | **75.47** | 68.27 | 59.75 |
| **B-AMOC** + $\mathcal{L}_{CE}$ | 85.90 | **45.17** | **12.02** | **0.14** | 76.78 | **64.29** | **50.99** | 75.38 | **68.64** | **60.97** |
| AMOC + AT | **84.48** | 50.87 | 16.85 | 0.26 | **77.07** | **67.28** | 56.00 | **76.14** | **70.45** | **64.43** |
| **B-AMOC** + AT | 83.80 | **50.89** | **17.81** | **0.38** | 76.35 | 66.79 | **56.16** | 75.44 | 69.78 | 64.29 |
| AMOC + **BAT** | **83.88** | 51.00 | 17.46 | 0.33 | **76.44** | 66.41 | 55.77 | **75.51** | **69.87** | 63.67 |
| **B-AMOC** + **BAT** | 83.40 | **51.08** | **18.47** | **0.37** | 75.97 | **66.45** | **56.11** | 75.15 | 69.48 | **63.83** |
| RoCL + $\mathcal{L}_{CE}$ | 83.69 | 38.49 | 8.73 | **0.66** | 65.98 | 61.12 | 44.47 | **68.03** | **67.59** | 60.42 |
| **B-RoCL** + $\mathcal{L}_{CE}$ | 85.06 | **40.44** | **9.54** | 0.63 | 65.37 | **62.86** | **47.42** | 66.42 | 66.63 | **63.67** |
| RoCL + AT | 79.65 | 47.35 | 16.33 | **0.36** | 67.33 | 65.18 | 53.38 | 68.20 | 68.17 | 65.58 |
| **B-RoCL** + AT | **79.69** | **49.36** | **17.41** | 0.33 | **67.58** | **66.15** | **54.64** | **68.21** | **68.54** | **66.68** |
| RoCL + **BAT** | 78.63 | 47.31 | 16.29 | 0.25 | **68.34** | 64.92 | 53.04 | **68.92** | **69.27** | 65.34 |
| **B-RoCL** + **BAT** | **79.69** | **49.33** | **17.22** | **0.38** | 67.59 | **66.03** | **54.47** | 68.38 | 68.43 | **66.73** |
| *Self-supervised + finetune* | | | | | | | | | | |
| 200 epochs: | | | | | | | | | | |
| AMOC + AF | 82.87 | 52.60 | **22.11** | 1.11 | 74.65 | 63.56 | 50.77 | 71.20 | 63.32 | **54.81** |
| **B-AMOC** + AF | **83.29** | **52.98** | 21.69 | **1.14** | **74.84** | **63.80** | **50.96** | **71.28** | **63.33** | 54.73 |
| AMOC + **B-AF** | 82.19 | 52.73 | 22.23 | **1.28** | 73.71 | 63.51 | **51.04** | 70.43 | 63.05 | **54.62** |
| **B-AMOC** + **B-AF** | 82.60 | **52.98** | **22.34** | 1.26 | **74.28** | 63.51 | 50.92 | **70.81** | 63.05 | 54.40 |
| 1000 epochs: | | | | | | | | | | |
| AMOC + AF | 83.28 | 52.82 | **22.04** | 1.12 | 74.95 | 63.87 | **51.38** | 71.79 | 63.83 | 55.13 |
| **B-AMOC** + AF | **84.00** | **53.08** | 21.74 | 1.09 | **75.44** | **64.65** | 51.20 | **71.95** | **64.20** | **55.33** |
| AMOC + **B-AF** | 81.85 | 52.62 | **22.51** | **1.38** | 73.77 | 63.21 | 50.99 | 70.82 | 63.16 | 54.55 |
| **B-AMOC** + **B-AF** | 82.76 | **53.07** | 22.17 | 1.32 | **74.63** | **64.11** | **51.49** | **71.17** | **63.83** | **55.30** |

Table 2: Results on Cifar-10 for self-supervised trained models. In the first part, the classification head was trained without adapting the pretrained features. In the second part, the parameters of the pretrained model were also adapted during training the classification head. $\mathcal{L}_{CE}$, AT, and BAT define, whether the classification head, and in case of finetuning the pretrained models, were trained on clean, adversarial, or with addition of pseudo adversarial inputs, respectively. A **B-** before the given self-supervised method indicates, that our proposed extension was applied. During training, the initial adversarial instances were created governed by $\ell_\infty$ with a strength of $8/255$.

tom of the right bar aside the grid), regarding the pixel value of the original input.

In all cases, we observe that $\ell_2$ and $\ell_1$ governed attacks tend to only slightly perturb the vast majority of pixel values while selecting a handful of pixels that are heavily perturbed. This is because the overall perturbation size for $\ell_2$ and $\ell_1$ is calculated over all pixels. Those attacks tend to spend their perturbation budget on the pixels, which seem to have the most impact on the classification. When the attack has the freedom to perturb each pixel independently, as is the case for $\ell_\infty$-attacks, the overall perturbation is larger. This also underlines the observation that clean trained models are more robust to $\ell_2$- and $\ell_1$-attacks. Particularly, when including additional input data during training, which introduces a larger variety of pixel value combinations. While at the same time, clean trained models are completely defenceless against attacks governed by $\ell_\infty$, which create manipulations that can not be covered by more clean data as the manipulations are too large and unnatural.

Considering these observations, we propose that BAT is less robust against small perturbations by $\ell_2$- and $\ell_1$-attacks, because it overfits to the observed perturbations, or more precisely the output vectors of such adversarial inputs, based on $\ell_\infty$ during training. In particular, since Cifar-10 includes only 50,000 samples.

This assumption is supported by the results for the supervised trained models, reported in Table 1, which used the additional 1 million samples for training. This additional data seem to prevent BAT from overfitting to the observed adversarial perturbation, as the variability in the input data, and thereby the variability in the pseudo adversarial instances, increases. This results in higher robustness to unseen attacks, compared to standard AT.

Another future step to prevent the potential overfitting would be to further investigate the manipulation distributions of $\ell_2$- and $\ell_1$-attacks, and in particular the distribution of their respective output vectors in the decision space. The gained insights could help to apply more sophisticated data augmentation in the decision space than the simple conditioned random noise we use here. Also, observing the distribution of clean sample output vectors could help to prevent pseudo adversarial inputs from jumping into a third classification area, as shown in Figure 2.

## Related Work

Recent works, e.g., (Madaan, Shin, and Hwang 2021; Rusak et al. 2020; Dong et al. 2020), propose to incorporate random noise into their techniques to increase the robustness of models against adversarial perturbations. To that end, (Madaan, Shin, and Hwang 2021) and (Rusak et al. 2020) employ some kind of generator, trained to create perturbations which are applied to the input vector, i.e., the image, before they are fed through the classification model. (Dong et al. 2020) as well, aim to model a distribution for each input which, when drawn from with very high probability returns an adversarial sample for the given input. Based on this learned adversarial distribution, the classification model itself is trained to minimize the expected loss over the adversarial distribution. In all these cases, the random manipulation is applied to the input vector, while we manipulate the output vector of a given adversarial sample. Because the works operate on different parts of the model, it should be possible to combine the techniques, to further increase adversarial robustness.

Regarding manipulating the output vector, mixup (Zhang et al. 2018) drew a lot of attention recently. (Lee, Lee, and Yoon 2020) took the idea of mixup and combined it with adversarial training, calling it Adversarial Vertex Mixup (AVM). Essentially, at first they create an adversarial sample and push it further in the adversarial direction to create the so called adversarial vertex. Then, instead of using two clean inputs as in the original mixup paper, they merge the initial clean sample and the adversarial vertex to form a new input. Since the clean sample and the adversarial vertex have the same label, the authors use some form of label smoothing function, e.g., by (Szegedy et al. 2016) to convert the one-hot encoded labels to a conditionally randomized distribution. Merging these two distributions give the new label for the mixup between the clean sample and the adversarial vertex. In contrast to this work, we create multiple adversarial instances, instead of one. AVM could be visualised as a line between the adversarial vertex and clean sample from which the new inputs are drawn. Our method creates a ball around the initial adversarial output vector from which multiple samples are drawn as new output vectors for training.

## Conclusion

Using data augmentation and larger datasets have shown to be supporting and sometimes even essential (Riquelme et al. 2021) to achieve better classification results and better generalisation. However, using these techniques does not yield robustness against adversarial manipulations. Instead, techniques like adversarial training are necessary to harden neural networks against unforeseen perturbations, which can fool the classification.

Since adversarial inputs are created in and defined by the output space, which ultimately leads to the decision of a model, we proposed to combine adversarial training with data augmentation in the output space, referring to as Broad Adversarial Training (BAT). We show, that already applying simply conditioned random noise to the output vectors of adversarial inputs, and thereby create multiple new pseudo adversarial inputs, can increase the robustness, and in some cases even the clean accuracy.

Extending standard Adversarial Training (AT) (Madry et al. 2017) to BAT for training on Cifar-10, increases the robustness against seen attacks by 0.55% for small perturbations and by 1.11% for larger perturbation size. On large unseen $\ell_2$-attacks the robust accuracy increases as well by 1.11%, and for large $\ell_1$-attacks by 0.29%. Increasing the clean data pool by another 1 million data points, using BAT increases the robust accuracy for all observed attacks between 0.12% and 0.79%, as well as the clean accuracy slightly by 0.05%. Similar results can also be reported for self-supervised learning, where using BAT can increase the robust and clean accuracy, as well.

## Acknowledgments

This work is funded by a DAAD-IFI program scholarship to NW and by a grant from the Center for Long-Term Cybersecurity at UC Berkeley. The authors thank Xudong Wang and Yunhui Guo for discussions, and Melanie Spindler for proofreading the paper.

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

## Appendix

### Optimization problems and loss functions

In Table 3 the respective loss functions and optimization problems used in training AMOC and RoCL are listed, with the explanation of the used symbols in the caption.

| | |
|---|---|
| Contrastive loss | $\mathcal{L}_{\mathrm{con},\theta}\left(x,\{x_{\mathrm{pos}}\},\{x_{\mathrm{neg}}\}\right)$ $:= -\log \dfrac{\sum_{\{f(x)_{\mathrm{pos}}\}}\exp\left(\mathrm{sim}\left(f(x),\{f(x)_{\mathrm{pos}}\}\right)/\tau\right)}{\sum_{\{f(x)_{\mathrm{pos}}\}}\exp\left(\mathrm{sim}\left(f(x),\{f(x)_{\mathrm{pos}}\}\right)/\tau\right)+\sum_{\{f(x)_{\mathrm{neg}}\}}\exp\left(\mathrm{sim}\left(f(x),\{f(x)_{\mathrm{neg}}\}\right)/\tau\right)}$ |
| RoCL PGD | $t_1\left(x\right)^{i+1}=\Pi_{B(t_1(x),\varepsilon)}\left(t_1\left(x\right)^i+\alpha\mathrm{sign}\left(\nabla_{t_1(x)^i}\mathcal{L}_{\mathrm{con},\theta}\left(t_1\left(x\right)^i,\{t_2\left(x\right)\},\{t_1\left(x\right)_{\mathrm{neg}}\}\right)\right)\right)$ |
| RoCL | $\underset{\theta}{arg\,min}E_{x\sim D}\left[\max_{\delta\in B(t_1(x),\varepsilon)}\mathcal{L}_{\mathrm{con},\theta}\left(t_1\left(x\right)+\delta,\{t_2\left(x\right)\},\{t_1\left(x\right)_{\mathrm{neg}}\}\right)\right]$ |
| MoCo Loss | $\mathcal{L}_{\mathrm{NCE}}=-\log\dfrac{\exp\left(q\cdot k_{\mathrm{pos}}/\tau\right)}{\exp\left(q\cdot k_{\mathrm{pos}}/\tau\right)+\sum_{k_{\mathrm{neg}}\in\mathcal{M}}\exp\left(q\cdot k_{\mathrm{neg}}/\tau\right)}$ |
| AMOC | $\min_{\theta_q,\theta_k}E_{x\in D}E_{t_1,t_2\in\mathcal{T}}\max_{\|\delta\|,\|\delta'\|\leq\varepsilon}\mathcal{L}\left(t_1\left(x\right)+\delta,t_2\left(x\right)+\delta',\mathcal{M}_{\mathrm{clean}},\mathcal{M}_{\mathrm{adv}}\right)$ |
| AMOC ACC | $\mathcal{L}_{\mathrm{ACC}}=\mathcal{L}_{\mathrm{NCE}}\left(f_q\left(t_1\left(x\right)+\delta;\mathrm{BN}_{\mathrm{adv}}\right),f_k\left(t_2\left(x\right);\mathrm{BN}_{\mathrm{clean}}\right),\mathcal{M}_{\mathrm{clean}}\right)$ |

Table 3: Loss functions and optimization problems defined for the used self-supervised adversarial training methods. In all formulations $x$ is a given clean sample, $f$ indicates the observed model, and $\delta$ is the adversarial perturbation. Also $t_1$ and $t_2$ define two different random transformation, while $\tau$ is some temperature hyperparameter. The contrastive loss is used within the RoCL framework, based on SimCLR, where $x_{\mathrm{pos}}$ and $x_{\mathrm{neg}}$ give the positive and negative samples, respectively. The similarity sim between two output vectors is calculated as the cosine similarity. To create an adversarial sample, the PGD equation (cf. Equation 1) is adapted by replacing the cross entropy loss with the contrastive loss. For the MoCo Loss, $q$ and $k$ represent the query and key encoder, respectively, and $\mathcal{M}$ is the used memory bank. While standard MoCo only defines one memory bank, within the AMOC framework the authors use two memory banks $\mathcal{M}_{\mathrm{clean}}$ and $\mathcal{M}_{\mathrm{adv}}$ to store the clean and adversarial historical samples. AMOC ACC is one specific loss function within the overall AMOC framework, for which the authors report good results, and which is therefore used in this study.

## Training parameters

In Table 4 all necessary hyperparameter for training the AMOC models are given, as well as for the supervised models/classification heads with clean, i.e., $\mathcal{L}_{\mathrm{CE}}$, and adversarial, i.e., AT, samples. The explanation for the certain abbreviations, e.g., which transformation are used under the term simclr, are explained in the caption. The same applies for the comprehensive list of hyperparameters for training RoCL and the respective classification heads, given in Table 5.

Further details:

- All experiments were run on NVIDIA GeForce GTX 1080.
- For RoCL training, we were not able to use the suggested batch size of 256 per GPU with our hardware.
- For RoCL we changed the projector to consist of 2, instead of 1, linear layers, followed by a normalization layer.
- Finetuning RoCL with only adversarial inputs led in our experiments to a classification accuracy of 10%. Using additional clean samples, we achieved a robust accuracy around 30%, which is 10% lower than the reported values, and would not be comparable to standard adversarial training. Therefore, RoCL + AF was excluded from our experiments.

## Additional time demand

In Table 6 the mean time required for one step of the indicated adversarial training method is listed. Further down, we split the time demand into the creation of the initial adversarial instance, which already takes up between 40.97 to 66.92% of the overall time. Calculating $\delta_{x',x}$ is only required once. Because for AT, the output vector of the clean sample is not calculated during training, the proportional time requirement is comparable large to the unsupervised methods, where the output vector is already calculated independent of our adaptation. Creating each pseudo adversarial input only adds a small portion, between 0.2% to 0.52% to the overall time demand per step. For RoCL the evaluation of the loss function furthermore takes up 84.96% of the time to create one pseudo adversarial instance. We explain this comparable large time demand by the fact that the RoCL framework implements the loss function itself, while AMOC uses the cross entropy loss provided by pytorch and does very limited own computation in context of the loss evaluation.

## Differences between the attacks

| | AMOC 200 | AMOC 1000 | AT head | AT | $\mathcal{L}_{\text{CE}}$ head | $\mathcal{L}_{\text{CE}}$ |
|---|---|---|---|---|---|---|
| GPU | 1 | 1 | 1 | 1 | 1 | 1 |
| optimizer | sgd | sgd | sgd | sgd | sgd | sgd |
| momentum | 0.9 | 0.9 | 0.9 | 0.9 | 0.9 | 0.9 |
| weight decay | 5e-4 | 5e-4 | 5e-4 | 5e-4 | 2e-4 | 2e-4 |
| learning rate | 0.1 | 0.1 | 0.1 | 0.1 | 0.1 | 0.1 |
| - decay | cosine | cosine | FC | TOTAL | FC | TOTAL |
| epochs | 200 | 1000 | 25 | 40 | 25 | 40 |
| warmup epochs | 10 | 10 | | | | |
| batch size | 256 | 256 | 128 | 128 | 128 | 128 |
| transform | simclr | simclr | default | default | default | default |
| attack: | | | | | | |
| type | $\ell_\infty$ | $\ell_\infty$ | $\ell_\infty$ | $\ell_\infty$ | | |
| $\varepsilon$ | 8/255 | 8/255 | 8/255 | 8/255 | | |
| step size | 2/255 | 2/255 | 2/255 | 2/255 | | |
| # steps | 5 | 5 | 10 | 10 | | |
| attack weight $\kappa$ | 0.5 | 0.5 | 1.0 | 1.0 | | |
| scatter operation: | | | | | | |
| $s_k$ | 10 | 10 | 10 | 10 | | |
| $\alpha_s$ | 0.25 | 0.25 | 2.5 | 2.5 | | |
| scatter decay | cosine | cosine | cosine | cosine | | |
| scatter weight $\lambda$ | 0.5 | 0.5 | 1.0 | 1.0 | | |
| MoCo specific: | | | | | | |
| dim_mlp | 512 | 512 | | | | |
| dim_head | 128 | 128 | | | | |
| $\tau$ | 0.2 | 0.2 | | | | |
| # samples in $\mathcal{M}_{\text{clean}}$ | 32768 | 32768 | | | | |
| # samples in $\mathcal{M}_{\text{adv}}$ | 32768 | 32768 | | | | |
| key encoder momentum | 0.999 | 0.999 | | | | |

Table 4: Full list of parameters for training AMOC, as well as the supervised models/classification heads with clean, i.e., $\mathcal{L}_{\text{CE}}$, and adversarial, i.e., AT, samples. FC is implemented to decaying the learning rate by a factor of 10 at epochs 10 and 15, while TOTAL reduces the learning rate by a factor of 10 at epochs 30 and 35. A default transformation is implemented as padding by 4, random resized cropping to 32, random horizontal flipping. SimCLR as transformation is composed of: random cropping of size 32, applying color jitter with a strength of 0.4 to the brightness, contrast, and saturation, while the hue is perturbed with strength 0.1, all with a probability of 0.8, random grayscale with a probability of 0.2, applying gaussian blur with a probability of 0.5, and random horizontal flipping. All inputs are converted to tensors, i.e., to the range [0, 1].

|  | RoCL | AT head | $\mathcal{L}_{CE}$ head |
|---|---|---|---|
| GPU | 2 | 1 | 1 |
| base optimizer | SGD | SGD | SGD |
| - momentum | 0.9 | 0.9 | 0.9 |
| - weight decay | 1e-6 | 5e-4 | 5e-4 |
| - learning rate | 0.1 | 0.2 | 0.2 |
| optimizer | LARS | | |
| - eps | 1e-8 | | |
| - trust_coeff | 0.001 | | |
| learning rate decay | cosine | | |
| warmup | GradualWarmUp | | |
| - lr multiplier | 15 | | |
| - warumup epochs | 10 | | |
| epochs | 1000 | 150 | 150 |
| batch size | 128 per GPU | 128 | 128 |
| transform | simclr | simclr | simclr |
| attack: | | | |
| type | $\ell_\infty$ | $\ell_\infty$ | |
| $\varepsilon$ | 0.0314 ($\approx 8/255$) | 0.0314 ($\approx 8/255$) | |
| step size | 0.007 ($\approx 2/255$) | 0.007 ($\approx 2/255$) | |
| # steps | 7 | 10 | |
| attack weight $\kappa$ | 1.0 | 1.0 | |
| scatter operation: | | | |
| $s_k$ | 10 | 10 | |
| $\alpha_s$ | 0.1 | 2.5 | |
| scatter decay | stepwise | cosine | |
| scatter weight $\lambda$ | 0.25 | 1.0 | |
| RoCL specific: | | | |
| $\tau$ | 0.5 | | |
| $\lambda_{RoCL}$ | 256 | | |

Table 5: Full list of parameters for training RoCL, as well as the supervised classification heads with clean, i.e., $\mathcal{L}_{CE}$, and adversarial, i.e., AT, samples. To train RoCL (Kim, Tack, and Hwang 2020) use the LARS (You, Gitman, and Ginsburg 2017) optimizer based on SGD with the given parameters. The initial learning rate is increased during the first 10 epochs by an overall factor of 15. Afterwards the learning rate is decayed by a cosine scheduler. Their input transformation is composed of: applying color jitter with a strength of 0.4 to the brightness, contrast, and saturation, while the hue is perturbed with strength 0.1, all with a probability of 0.8, random grayscale with a probability of 0.2, random horizontal flipping, and random resized cropping of size 32. All inputs are converted to tensors, i.e., to the range [0, 1].

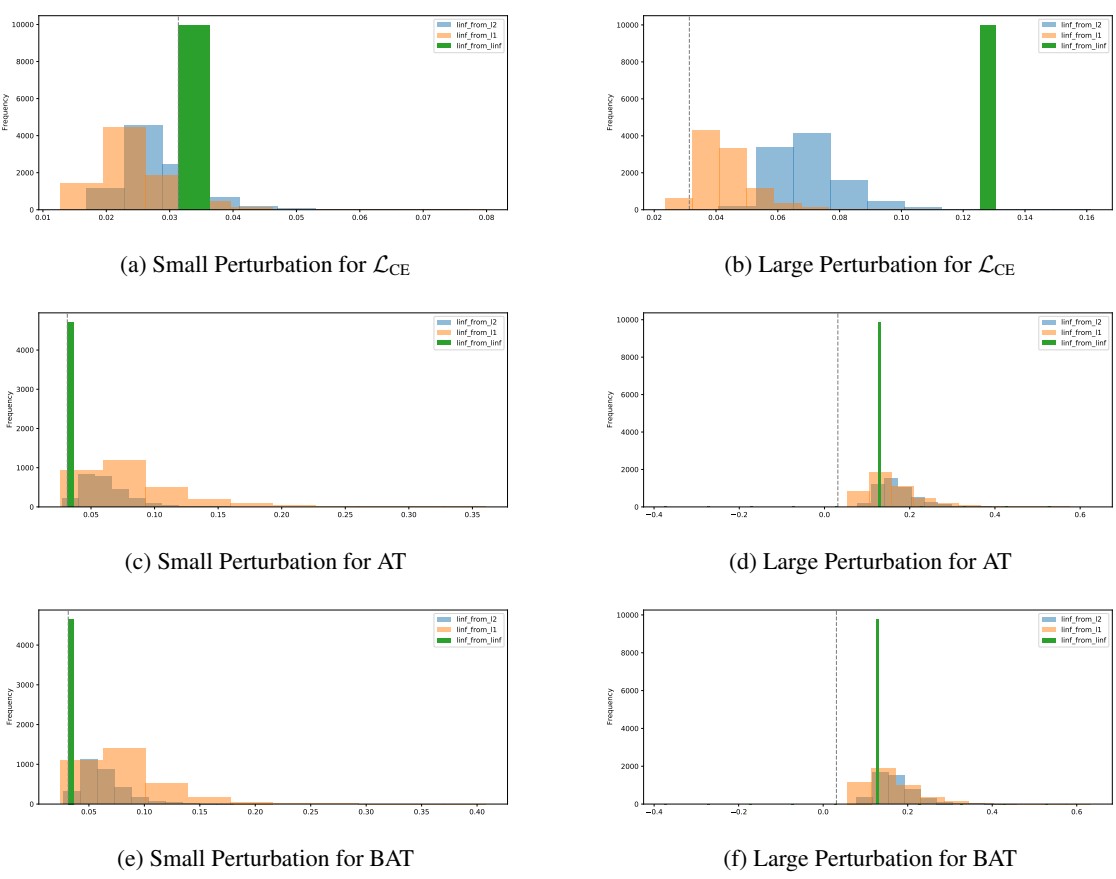

Figure 3: Distribution of perturbation size, measured regarding $\ell_\infty$, for attacks governed by $\ell_2$ (blue), $\ell_1$ (orange), and $\ell_\infty$ (green) on the indicated trained model. The gray line indicates a perturbation size of $\ell_\infty = 8/255$, giving a landmark for seen adversarial inputs during adversarial training.

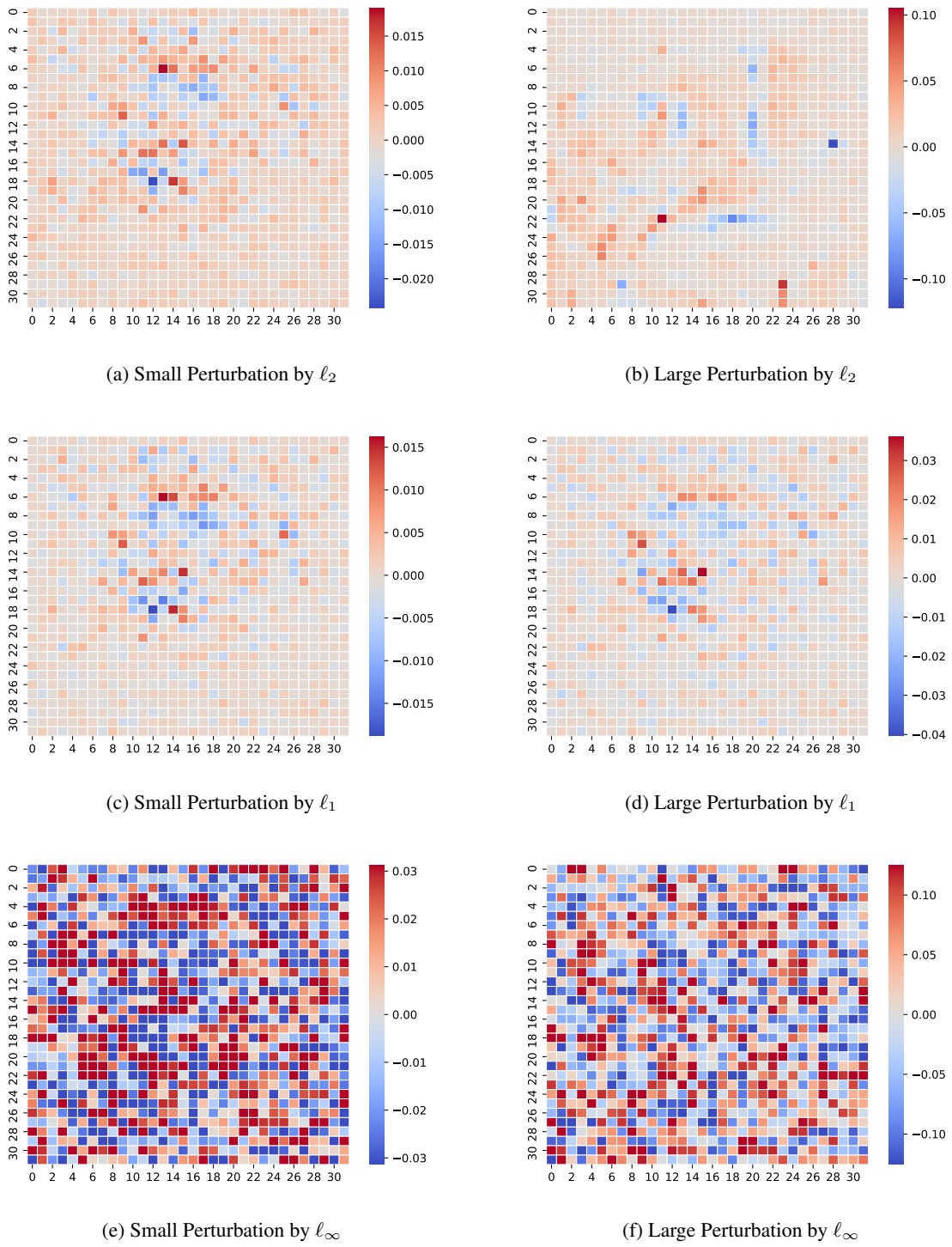

(a) Small Perturbation by $\ell_2$

(b) Large Perturbation by $\ell_2$

(c) Small Perturbation by $\ell_1$

(d) Large Perturbation by $\ell_1$

(e) Small Perturbation by $\ell_\infty$

(f) Large Perturbation by $\ell_\infty$

Figure 4: Perturbation for each pixel governed by $\ell_2$ (top), $\ell_1$ (middle), and $\ell_\infty$ (bottom), measured regarding $\ell_\infty$ on a $\mathcal{L}_{\text{CE}}$ trained model.

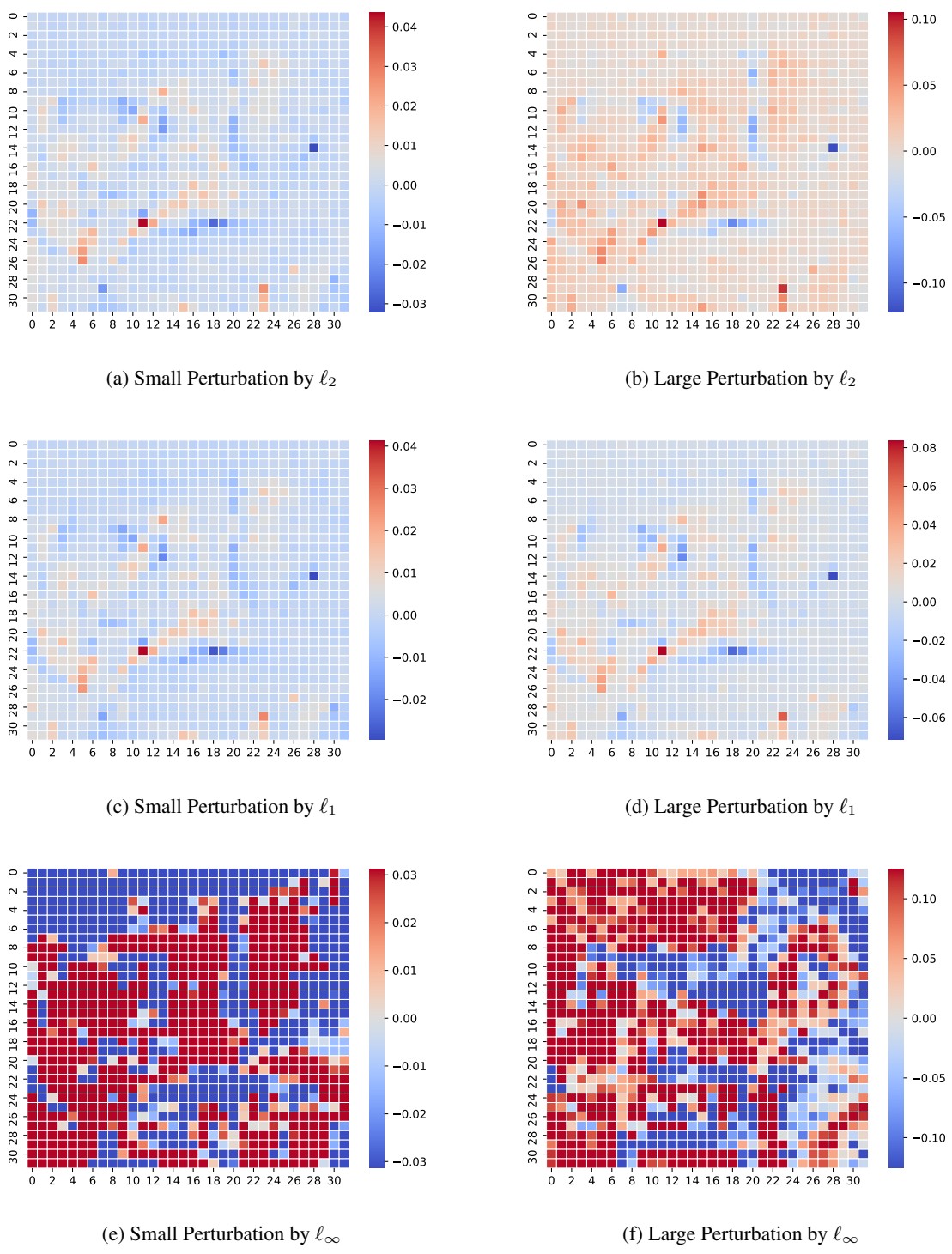

(a) Small Perturbation by $\ell_2$

(b) Large Perturbation by $\ell_2$

(c) Small Perturbation by $\ell_1$

(d) Large Perturbation by $\ell_1$

(e) Small Perturbation by $\ell_\infty$

(f) Large Perturbation by $\ell_\infty$

Figure 5: Perturbation for each pixel governed by $\ell_2$ (top), $\ell_1$ (middle), and $\ell_\infty$ (bottom), measured regarding $\ell_\infty$ on a PGD adversarial trained model.

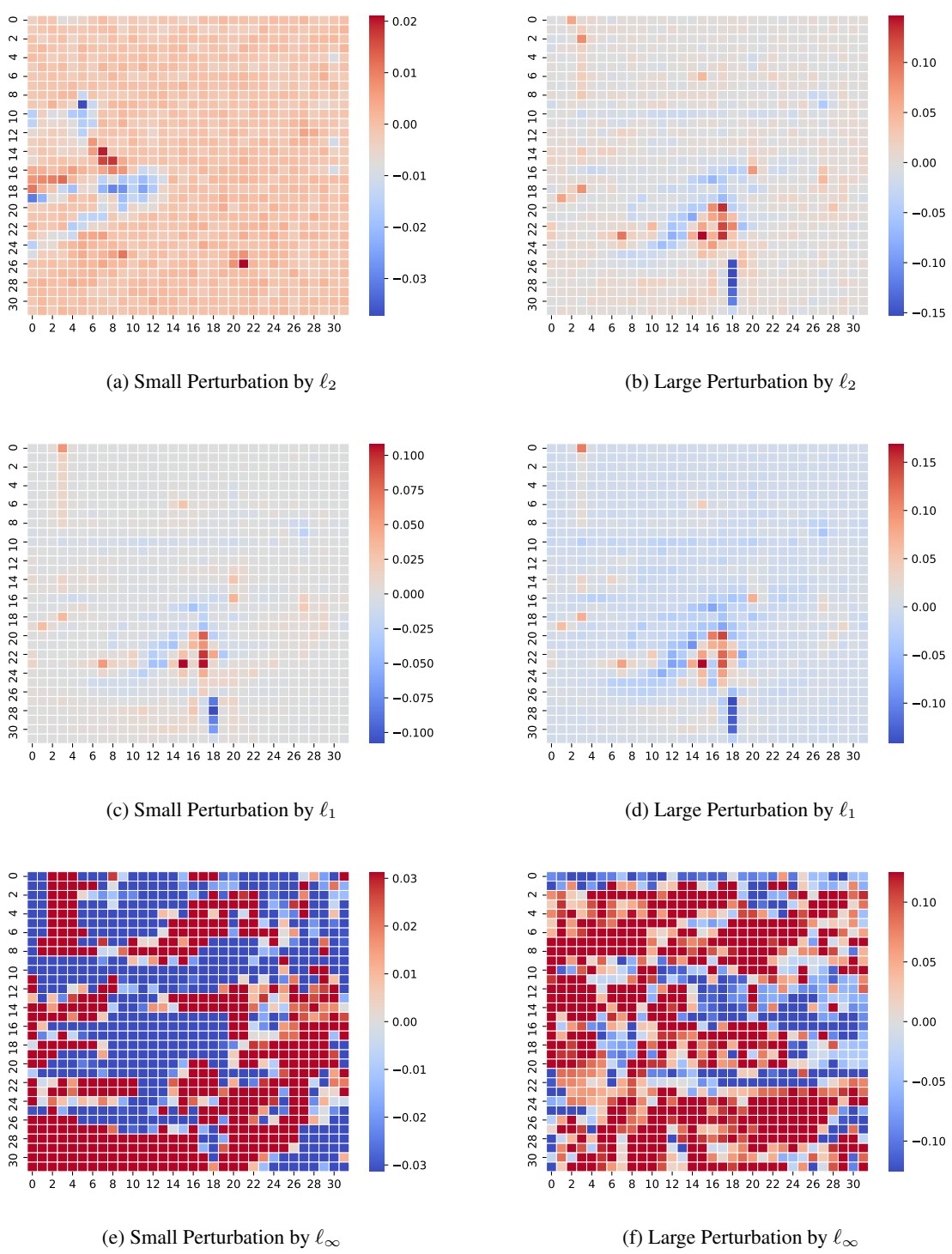

(a) Small Perturbation by $\ell_2$

(b) Large Perturbation by $\ell_2$

(c) Small Perturbation by $\ell_1$

(d) Large Perturbation by $\ell_1$

(e) Small Perturbation by $\ell_\infty$

(f) Large Perturbation by $\ell_\infty$

Figure 6: Perturbation for each pixel governed by $\ell_2$ (top), $\ell_1$ (middle), and $\ell_\infty$ (bottom), measured regarding $\ell_\infty$ on our proposed broad adversarial trained model.

| operation | mean in ms | std in ms | % of overall time |
|---|---|---|---|
| **AT:** | | | |
| overall time per step | 777.55 | 30.37 | |
| create initial adversarial input | 429.14 | 16.61 | 55.19 |
| calculate $\delta_{x',x}$ | 43.65 | 4.03 | 5.61 |
| create one pseudo adversarial input | 1.56 | 3.68 | 0.20 |
| calculate the loss within scattering | 0.09 | 0.02 | 0.01 |
| **AMOC:** | | | |
| overall time per epoch | 1180.50 | 59.58 | |
| create initial adversarial input | 483.61 | 26.72 | 40.97 |
| calculate $\delta_{x',x}$ | 0.10 | 0.01 | 0.01 |
| create one pseudo adversarial input | 3.65 | 7.23 | 0.31 |
| calculate the loss within scattering | 0.26 | 0.19 | 0.02 |
| **RoCL:** | | | |
| overall time per epoch | 1342.06 | 52.27 | |
| create initial adversarial input | 898.09 | 35.55 | 66.92 |
| calculate $\delta_{x',x}$ | 0.81 | 0.14 | 0.06 |
| create one pseudo adversarial input | 7.04 | 1.25 | 0.52 |
| calculate the loss within scattering | 5.98 | 1.22 | 0.45 |

Table 6: Time demand for different operations during BAT given in ms. For each method we list the overall mean time and standard deviation for one step, as well as the time required to calculate the initial adversarial input. The overall scatter operation is split into calculating $\delta_{x',x}$, which is only performed once, and the creation of one pseudo adversarial input. In particular, we also list the time required to evaluate the loss function for the created pseudo adversarial instance.