# OpenReview forum: "Broad Adversarial Training with Data Augmentation in the Output Space"
_AAAI.org/2022/Workshop/AdvML — AAAI-22 AdvML Workshop LongPaper_

### Official Review · Reviewer_CcYk · 2021-11-27
**Review for the paper "Broad Adversarial Training with Data Augmentation in the Output Space"**

**Rating:** 7
**Confidence:** 4

**Review:**

This paper proposes an interesting approach for adversarial training, that the data augmentation is applied to the output space.

Advantages:
- Figure 1 is clear to show the motivation of this paper, while is also reasonable.
- The proposed technique is simple and general for various adversarial training paradigms, including supervised learning, self-supervised learning.
- The results are clear to show the effectiveness.

I have some suggestions to further improve this paper:
- The section number is missing in the paper, e.g., in the first paragraph in Method, "Multiple approaches for adversarial training are outlined in Section ."
- The authors are encouraged to conduct experiments using AutoAttack.

Overall, this is a good paper with sufficient contribution to the field. Thus I recommend acceptance.

---

### Decision · Program_Chairs · 2021-12-01

**Decision:**

Accept (Long Paper)

**Comment:**

The reviewer agrees to accept this paper. Please consider the suggestions in the camera-ready version.